# Applying Zinc Nutrient Reference Values as Proposed by Different Authorities Results in Large Differences in the Estimated Prevalence of Inadequate Zinc Intake by Young Children and Women and in Cameroon

**DOI:** 10.3390/nu14040883

**Published:** 2022-02-19

**Authors:** Demewoz Haile, Kenneth H. Brown, Christine M. McDonald, Hanqi Luo, Michael Jarvis, Ismael Teta, Alex Ndjebayi, Guintang Assiene Jules Martial, Stephen A. Vosti, Reina Engle-Stone

**Affiliations:** 1Department of Nutrition, University of California, Davis, CA 95616, USA; khbrown@ucdavis.edu (K.H.B.); christine.mcdonald@ucsf.edu (C.M.M.); hanqi.luo@emory.edu (H.L.); renglestone@ucdavis.edu (R.E.-S.); 2Institute for Global Nutrition, University of California, Davis, CA 95616, USA; mjarvis@ucdavis.edu (M.J.); vosti@primal.ucdavis.edu (S.A.V.); 3Departments of Pediatrics, and Epidemiology and Biostatistics, School of Medicine, University of California, San Francisco, CA 94143, USA; 4Hubert Department of Global Health, Rollins School of Public Health, Emory University, Atlanta, GA 30322, USA; 5Helen Keller International, Yaoundé 1771, Cameroon; iteta@hki.org (I.T.); andjebayi@hki.org (A.N.); 6Nutrition International, Yaoundé 1771, Cameroon; jassiene@nutritionintl.org; 7Department of Agricultural and Resource Economics, University of California, Davis, CA 95616, USA

**Keywords:** dietary assessment, modeling, zinc, fortification, children, women, Cameroon

## Abstract

Nutrient reference values (NRVs) for zinc set by several expert groups differ widely and may affect the predicted prevalence of inadequate zinc intake. We examined this possibility using NRVs published by four different authorities and nationally representative dietary intake data collected among children aged 12–59 months and women in Cameroon. Usual zinc intake was estimated from 24 h recall data using the National Cancer Institute method. Prevalences of total zinc intake below the dietary requirement and of “absorbable zinc intake” below the physiological requirement were estimated using NRVs published by the World Health Organization (WHO), US Institute of Medicine (IOM), International Zinc Nutrition Consultative Group (IZiNCG), and European Food Safety Authority (EFSA). The prevalence of inadequate zinc intake ranged from 10% (IZiNCG—physiological requirement, 95% CI 7–13%) to 81% (EFSA—physiological requirement, 95% CI 78–84%) among children and 9% (WHO—physiological requirement, 95% CI 8–11.0%) to 94% (IOM—physiological requirement, 95% CI 92–95%) among women These differences in the prevalence of inadequate intake translated into sizeable differences in the predicted benefit and cost-effectiveness of zinc fortification programs. Depending on the NRVs applied, assessments differ regarding the need for and design of zinc fortification programs. Efforts are needed to harmonize NRVs for zinc.

## 1. Introduction

Adequate zinc nutrition is necessary for the optimal health and physical growth of children and for normal pregnancy outcomes [1,2,3,4]. Zinc deficiency is a global nutritional problem, particularly among children and women living in low- and middle-income countries (LMICs), where diets are often monotonous and based primarily on cereals or tubers, with limited zinc bioavailability [5]. Among 25 LMICs that have nationally representative data on plasma zinc concentrations (PZCs), 23 countries had a zinc deficiency prevalence >20% among young children, women of reproductive age, or both [5]. In sub-Saharan Africa, the availability of nationally representative zinc biomarker data is limited, but in five of six countries with nationally representative PZC data, more than 40% of children have low PZC [5,6,7,8].

Dietary diversification or modification is one strategy to prevent deficiencies of zinc and other micronutrients, but the necessary economic and behavior change interventions are complex and require a relatively long timeframe to implement [9]. Other interventions, such as preventive zinc supplementation and fortification, have been recommended as complementary strategies to address zinc deficiency, particularly among young children and women of reproductive age [10]. Previous research has shown that zinc fortification improves plasma zinc concentration and other functional outcomes [11,12,13], but evidence of program effectiveness is limited. The 2021 *Lancet* Series on Maternal and Child Undernutrition Progress recommended large-scale food fortification (LSFF) as a strategy with “strong evidence for implementation” for preventing micronutrient deficiencies, including zinc [14]. Currently, about 30 LMICs have regulations for either mandatory or voluntary zinc fortification of staple foods such as wheat and/or maize flour [5].

Fortification programs require careful planning to determine the optimal level of each micronutrient fortificant necessary to reduce the prevalence of dietary inadequacy without increasing intake above the tolerable upper intake level (UL) [15]. Simulation studies can provide insights about the potential contribution of food fortification to the reduction in nutrient inadequacy, which can be used both for political advocacy and program planning by providing an efficient way of comparing hypothetical fortification scenarios [16]. Historically, the ability to undertake such studies has been limited by the availability of appropriate dietary intake data (i.e., 24 h dietary recalls or the equivalent from a representative sample of the target population). The use of modeling may increase as new tools to facilitate the collection of dietary intake data become available and more countries collect these data [17,18]. Consideration of predicted program benefits and costs allows decision-makers to select strategies that address health and nutrition priorities while meeting budget constraints [19,20,21]. Because the usefulness of simulation modeling studies depends on the validity of inputs and assumptions used, guidance on best practices for these studies will help generate results that are accurate and reliable to inform programmatic decisions.

Information on the estimated physiological requirements and dietary reference values for a nutrient is critical for determining the prevalence of inadequate micronutrient intake and simulating the impact of hypothetical fortification scenarios. Several expert groups, including the World Health Organization (WHO), the US Institute of Medicine (IOM), the International Zinc Nutrition Consultative Group (IZiNCG), and the European Food Safety Authority (EFSA), have estimated physiological zinc requirements and dietary reference values for zinc, but these published nutrient reference values (NRVs) are inconsistent. The differences in physiological requirements reported by these expert groups are mainly due to differences in the data sources, methods used to estimate endogenous zinc losses, and reference body weights that were applied [22], as described below in more detail. Some of the differences are substantial. For example, the physiological requirement for children aged 1–3 years set by EFSA is 2-fold higher than that published by IZiNCG (mainly attributable to differences in figures used for endogenous zinc losses when estimating requirements). There are also differences in the level of fractional zinc absorption applied by these expert groups to convert the physiological requirements to dietary requirements. Further, algorithms to predict zinc absorption based on individual characteristics and dietary components are now available for adults [23] and children [24]; thus, the resulting estimates of absorbable zinc may be compared directly to the physiological requirement. However, the fractional zinc absorption predicted by these algorithms may differ from the estimates used to derive dietary requirements.

There is no clear guidance about which of the expert groups’ recommendations and which method of accounting for absorption are the most appropriate for dietary zinc assessment in LMICs. The implications of using different reference values to estimate the prevalence of inadequate zinc intake and predict the effect of zinc fortification on dietary zinc adequacy are uncertain. Variations in the estimated benefit of a fortification program when different nutrient reference values (NRVs) are used in simulation modeling may lead to skepticism about the benefit of proposed policies or programs and ultimately impede efforts to improve zinc nutrition globally.

This study aimed to estimate the prevalence of inadequate zinc intake and the predicted impact of zinc fortification using four different sets of zinc reference values (WHO, IOM, IZiNCG, and EFSA) as well as different methods of accounting for zinc absorption (i.e., comparing total intakes to dietary requirements or comparing absorbable zinc intake to physiological requirements) among young children and women of reproductive age using nationally representative data from Cameroon. We combined these estimates with program cost estimates to assess how these differences might translate to estimated cost-effectiveness. Finally, we evaluated whether the results generated using these varying methods differ in ways that might affect policy or program decisions.

## 2. Materials and Methods

### 2.1. Study Settings and Design

The current analyses relied on data from a national survey conducted in Cameroon in 2009 before the introduction of mandatory zinc fortification of wheat flour. The survey used a stratified, multistage cluster design and was conducted in three geographic strata of Cameroon: North, South, and the major metropolitan centers of Yaoundé and Douala. The clusters drawn from the North and South represented a mix of smaller urban and rural locations. From each stratum, 30 clusters were randomly selected, and then 10 households (i.e., 10 women and 10 children) were selected per cluster. A multiple-pass 24 h dietary recall interview was used to collect dietary intake data from a total of 883 children 12–59 mo of age (with repeated assessments on a randomly selected subsample of 66 children) and 912 women of reproductive age (with repeated assessment on 72 randomly selected women). Individuals with severe illness in the 72 h prior to data collection were excluded [25]. We constructed food composition tables using nutrient values from the Nutrition Coordinating Center Nutrient Database for Standard Reference [26], supplemented with values from the United States Department of Agriculture, a food composition table from Uganda [27], and the nutritional composition of commonly consumed dishes from Cameroon [28]. From the 24 h dietary recall, we estimated intake of total zinc, “absorbable zinc” (as described below), wheat equivalents (grams of wheat flour derived from various foods, such as bread or biscuits), and bouillon cubes among children and women. Those individuals with implausible energy intake (group-specific mean + 3 SD) were excluded from the analysis. Details of the sampling approach, sample size, dietary data collection methods, and calculation of reported nutrient intakes and fortifiable foods have been presented elsewhere [29,30].

### 2.2. Dietary Modeling

#### 2.2.1. Zinc NRVs

The EAR cut-point method was used to estimate the prevalence of inadequate zinc intake based on WHO, IOM, IZINCG, and EFSA zinc NRVs, including both dietary requirements and physiological requirements. All four groups used the factorial approach as the basis to derive physiological requirements, which are the amount of absorbed zinc required to offset all obligatory zinc losses plus any additional zinc needed to support physical growth, pregnancy, or lactation. Each of the expert groups then derived an estimated average requirement (referred to as the “dietary requirement”) from the physiological requirement by adjusting for the estimated absorption of zinc from the habitual diet at the level of zinc intake estimated to just satisfy the physiological requirement. In the present paper, we use the terms “dietary requirement” and “estimated average requirement (EAR)” interchangeably. The methods and assumptions used by the different expert groups to estimate physiological and dietary zinc requirements have been reviewed in more detail elsewhere [22,31]. Expert group deliberations differed with respect to the data sources and method used to estimate endogenous fecal zinc losses (EFZ), non-intestinal zinc loss (from urine, integument, semen, and menses), and reference body weight [22].

Table 1 summarizes the dietary and physiological requirements and ULs published by each expert group. The differences in estimated EFZ losses among these expert groups played a major role in the substantial variation in physiological zinc requirements established by each expert group [22]. Appendix A summarizes differences in the methods of estimating total endogenous zinc losses and fractional absorption of zinc among expert groups. Below, we briefly summarize the approaches taken and major assumptions applied by each group.

WHO NRVs for zinc were first established in 1996 and updated in 2004 [32]. WHO estimated EFZ based on two studies of fecal and urinary zinc excretion with diets very low in zinc [33,34]. Because these studies did not use isotopic tracers, it was not possible to distinguish between unabsorbed dietary zinc and EFZ. Moreover, because EFZ increases in relation to absorbed zinc, data from these studies do not reflect EFZ under conditions of zinc balance. For these reasons, WHO developed a two-step process considering both “basal requirements” and “normative physiological requirements”. Basal requirements refer to the amounts needed to balance the physiological needs of individuals who are fully adapted to low zinc intakes. The ”normative physiological requirement” accounts for the fact that zinc absorption must be about 40% greater to balance fecal and urinary losses in individuals who are not adapted to low intakes [35]. Thus, WHO inflated EFZ estimates by 40% to account for the reduced excretion of zinc with very restricted zinc intakes. Similarly, the urinary zinc losses estimated from these studies were inflated by 40% to account for the reduced urinary excretion observed with very low zinc intakes. Integumentary zinc losses were estimated based on a single study of adult men and extrapolated to women. WHO did not estimate zinc losses via semen or menstrual flow. WHO extrapolated zinc requirements for children from those for adults based on metabolic rate and established three dietary zinc reference values depending on the dietary phytate/zinc molar ratio [32] (Appendix A).

IOM published zinc reference values in 2001. The EFZ was estimated based on radioactive or stable isotope tracer data generated from 10 whole-day diet studies conducted in North America and Europe [36]. Linear regression was used to quantify the relationship between EFZ and total absorbed zinc (TAZ). Urinary zinc losses (0.63 mg/d) were based on 10 studies of men and women whose zinc intakes were within the range in which urinary zinc excretion is constant. IOM estimated integumentary zinc losses based on a single study in adult males and extrapolated the results to women based on body surface area. Zinc losses via semen or menstrual flow were each assumed to be 0.1 mg/day. IOM estimated the fractional absorption of zinc to be 30% for children and 27% for nonpregnant, nonlactating women [36]. IZiNCG published zinc NRVs in 2004. IZiNCG estimated EFZ losses based on 19 tracer studies among men and women, irrespective of age and nationality, who consumed mixed diets. The relationship between total absorbed zinc (TAZ) and EFZ was examined separately for men and women using linear regression and weighted by the sample size. To examine the relationship between the mean amount of absorbed zinc intake and total dietary zinc intake (i.e., fractional absorption of zinc), IZiNCG used whole-day diet studies of both men and women and excluded semi-purified formula diets and zinc supplements. IZiNCG adopted the same figures as IOM for zinc losses in urine, integument, and semen but assumed that losses in menstrual fluid are negligible (~5 µg/day). IZiNCG calculated various dietary requirements assuming fractional zinc absorption of 18–34%, depending on gender, and classified the bioavailability of the diet using the phytate/zinc molar ratio [22].

The EFSA zinc NRVs were published in 2014 [37]. This committee included almost all studies that IZiNCG included but applied a different regression approach from IZiNCG in estimating the physiological requirements. Specifically, EFSA used multiple regression analysis to examine the relationship between TAZ and EFZ. Sex differences were accounted for by including body weight as a covariate in the regression model. EFSA estimated integument and sweat losses (0.5 mg/day) based on studies conducted among men and extrapolated these results for women (0.30 mg/day) using a female-to-male ratio of sweat zinc losses and whole-body sweat rates. EFSA assumed zinc losses in semen and menses of 0.1 and 0.01 mg/day, respectively, and calculated urinary zinc losses of 0.54 mg/day for men and 0.32 mg/day for women. EFSA examined the relationship between absorbed zinc and total zinc intake using a trivariate response saturation model to derive dietary requirements based on whole-day diet studies and considering both zinc and phytate intakes. A total of 72 mean data points reflecting 650 individual measurements were included. Semi-purified diets and zinc supplements were excluded from the analysis. EFSA published average dietary requirements for adults based on four phytate levels (300, 600, 900, and 1200 mg/day) [36]. For children, EFSA adopted the fractional zinc absorption level established by IOM to derive the dietary requirements.

Hambidge et al. (2011) reported errors in the estimated physiological zinc requirement for adults proposed by IOM and IZiNCG [31]. In particular, the IOM zinc reference values used an erroneous value for menstrual losses (0.1 mg Zn/day instead of 0.01 mg Zn/day), and IZiNCG included two incorrect values for EFZ from one study by mistakenly considering the calculated absorbed isotope secreted into the intestine as EFZ values. In addition, the IZiNCG approach of applying a sample-size-weighted regression in examining the relation of EFZ with TAZ was criticized because the sample size and the variance were not correlated in the expected direction. After these errors were corrected for women, the IOM zinc physiological requirement values decreased from 3.20 mg Zn/day to 2.97 mg Zn/day, and the IZiNCG estimates increased from 1.86 to 2.87 mg Zn/day. After correcting for the foregoing errors, the differences in IOM, IZiNCG, and EFSA physiological requirements became smaller, but other minor differences still remained. In this study, we applied both the original IOM and IZiNCG physiological requirement values as well as the corrected ones to estimate the prevalence of inadequate zinc intake among women.

WHO, IOM, IZiNCG, and EFSA established the respective ULs based on studies reporting the adverse effect of high zinc intake on copper biomarkers. WHO estimated a UL of 45 mg/day for adults based on studies that reported an adverse effect of 50 mg zinc/day on copper status. This value was extrapolated to other population groups in relation to basal metabolic rate [38]. IOM established a UL for adults based on the lowest observed adverse effect level from a single study with a total intake of 60 mg Zn/d (50 mg Zn/d from supplement and 10 mg/d from diet), while that for children was based on the no observed adverse effect level (NOAEL) from a zinc-fortified infant formula containing 5.8 mg Zn/L [36,39]. IZiNCG estimated the UL for children based on a NOAEL reported from a single study conducted in Indonesia in which infants aged 6 months were supplemented with 10 mg zinc or a placebo for 6 months, with no apparent change in plasma copper levels [40]. For women, IZiNCG published the same UL reference values as IOM [22]. The most recent EFSA reference values did not establish a UL for any of the life stages [37], but EFSA subsequently published a NOAEL for zinc in 2006 [41] based on a review of studies that showed no adverse effect on a wide range of copper indicators at 50 mg/day zinc intake among adults. The UL for children and adolescents was extrapolated from adults based on body surface area. In the present study, we estimated the prevalence of total zinc intake above the UL based on the reference values proposed by each of the four expert groups (Table 1).

#### 2.2.2. Zinc Absorption

As noted above, dietary reference intakes are derived by applying a value for fractional zinc absorption, overall or by type of diet. Based on phytate/zinc molar ratios, both children’s and women’s diets in Cameroon would be considered moderately bioavailable for zinc (phytate-to-zinc molar ratio: 5–15) [32]. Another approach for assessing dietary zinc adequacy is to apply an algorithm to estimate absorbable dietary zinc and to compare these values with the physiological requirements. For women, we used Miller and colleagues’ updated trivariate saturation response model published in EFSA’s report [23,37,42]. In the algorithm, the independent variables included were total phytate intake and total dietary zinc intake [23]. For this study, we used total phytate intake adjusted for fermentation of cereal products, as described elsewhere [43]. For young children, we applied the zinc absorption algorithm published by Miller and colleagues in 2015 [24]. The authors found that age and total dietary zinc intake were the only predictors of total absorbed zinc and that dietary phytate intake was not related to zinc absorption in this age group [24].

#### 2.2.3. Simulation of Effects of Zinc Fortification

We simulated the effects of zinc fortification of different vehicles, alone or combined, to assess the extent to which the conclusions about predicted impact would change with different nutrient reference values. In Cameroon, wheat flour and bouillon cubes were identified as potential vehicles for zinc fortification in the 2009 national survey [30]. Because of the different consumption patterns of these two vehicles, we assessed whether the application of different reference values would be more apparent depending on the vehicle fortified. In Cameroon, the reach of wheat flour was limited (48% of individuals consumed products containing wheat flour on the previous day), and the amount consumed per individual was relatively large (compared with bouillon cubes) and highly positively skewed. By contrast, bouillon cube was more widely consumed (88% on the previous day), and the amount consumed was relatively small and less variable among individuals.

For countries such as Cameroon, where the average per capita wheat flour availability is less than 75 g/day [30], the zinc fortification level recommended by WHO for the low wheat flour extraction rate is 95 mg zinc/kg flour [44], which we used for these simulations. Mandatory fortification of wheat flour with zinc has been part of Cameroon’s national fortification strategy since August 2011. The fortification levels selected for bouillon cubes were hypothetical values based on multiples of the current food industry practice for iron fortification of bouillon cubes (voluntary fortification at ~0.6 mg iron/g bouillon cube). The technical feasibility of the simulated range of zinc concentrations in bouillon cubes (0.6, 1.2, 1.8, 3.0, and 5.0 mg zinc/g bouillon cube) would need to be confirmed. We considered this range of concentrations of interest for the purpose of the comparisons of zinc NRVs. In this study, we simulated the impact of wheat flour fortification and various levels of zinc fortification of bouillon cubes on the prevalence of inadequate zinc intake using each of the NRVs. We also evaluated the impact of the various zinc fortification levels of bouillon cubes on the prevalence of inadequate zinc intake in the presence of the wheat flour fortification program (95 mg zinc/kg wheat flour).

#### 2.2.4. Estimation of Usual Intake

We applied the National Cancer Institute (NCI) method to estimate the distributions of usual zinc intake using the University of California, Davis/NCI SIMPLE macro [17]. The SIMPLE macro allows the application of population-specific NRVs (i.e., different values for children aged 1–3 years vs. 4–5 years and for pregnant vs. nonpregnant vs. lactating women as published by each expert group, rather than using a single weighted average). To estimate usual absorbable zinc intake, we first applied the zinc absorption algorithm to estimate the absorbable zinc intake for children and women on the day of the dietary assessment. Then, we used this estimated absorbable zinc intake in the NCI amount-only model to estimate the distributions of usual absorbable zinc intakes. In addition, total zinc intake was used in the NCI amount-only model to estimate usual total zinc intake because there were only a few days with observed zinc intakes of zero. The estimated ratio of within-person to between-person variance was checked, and observations (14 child-days) that inflated the ratio above 10 were excluded, as suggested by Davis et al. (2019) [45]. For children, usual intake was adjusted for age, sex, interviewer ID, sequence of interviews (i.e., first interview compared with subsequent interview), use of translator in the dietary interview, weekend (binary variable indicating weekend day compared with weekday), breastfeeding status, maternal education (secondary/higher, primary, or no formal education), socioeconomic status, and macro-region (stratum) [46]. Similarly, for women, usual zinc intake was adjusted for maternal age, interviewer ID, sequence of interviews, use of translator, weekend, maternal education (secondary/higher, primary, or no formal education), socioeconomic status, macro-region (stratum), and physiologic status (nonpregnant/nonlactating, pregnant, lactating). A study by Brown et al. estimated that total zinc intake from breast milk for partially breastfed infants aged 12–17 mo was 0.29 mg/d [47]. Assuming that the bioavailability of zinc from breast milk is 50%, for children who were reportedly still breastfed at the time of the survey, we assumed an additional absorbable zinc intake of 0.145 mg/day from breast milk. Zinc intake from breast milk was added to the usual intake distribution using the “shrink then add approach” [48]. In the “shrink then add approach”, zinc intakes from food sources are first processed through the NCI amount-only model to generate a representative sample of the modeled usual zinc intakes from food sources, and then the estimated zinc intake from breast milk is added to each modeled intake of breastfeeding children to produce a representative sample of usual zinc intakes (total zinc intake or absorbable zinc intake). In the shrink then add approach, we used total daily zinc intake from breast milk (0.29 mg) for simulation scenarios that aimed to estimate the usual total dietary zinc intake distributions, while absorbable zinc intake from breast milk (0.145 mg) was used for simulation scenarios based on estimated usual absorbable zinc distributions. The dietary requirements of the respective expert groups were used to estimate the prevalence of inadequate zinc intake from the total usual zinc intake, while the physiological requirements were used to estimate the prevalence of inadequate intake from the usual absorbable zinc intake. Because the survey employed a complex sampling design, we used Fay’s modified balanced repeated replication (BRR) procedure to obtain appropriate standard errors (SE) [46]. The prevalence of inadequate zinc intake based on each expert group’s dietary and physiological requirements is presented with standard error (±SE). We also report the mean ± SE and median (25th percentile, 75th percentile) of usual zinc intake (total zinc intake and absorbable zinc intake), phytate, and the phytate/zinc molar ratio.

#### 2.2.5. Fortification Program Costs and Cost-Effectiveness

We included cost data to explore the effects of using different NRVs to estimate the prevalence of inadequate intake on the estimated cost-effectiveness of two zinc fortification programs: zinc-fortified wheat flour (95 mg zinc/kg flour) and zinc-fortified bouillon cubes (5 mg zinc/g bouillon cube)**.** The costs of zinc fortification programs are grouped into two broad cost categories: establishment costs and operating costs. For ongoing programs (e.g., wheat flour fortification, which has been mandatory since 2011), establishment costs include only the marginal costs of adding zinc to the premix. For new programs, such as the fortification of bouillon cubes, which are currently hypothetical, establishment costs include setting official standards for fortification, organizing private sector investments needed to be able to fortify products, and upgrading public sector program monitoring to accommodate new products, new tests, etc.

An activity-based cost analysis was conducted to determine the costs of the establishment, operation, and monitoring and evaluation of zinc fortification. We assumed that 100% of bouillon cubes and wheat flour, whether produced domestically or imported, are fortified with zinc oxide; the same assumption was made when estimating the nutritional benefits, and hence, both the benefits and the costs represent upper-bound estimates of program performance. The cost-effectiveness was calculated for a 10-year planning horizon of zinc fortification programs; calculations were performed separately for children and women. Using the UN world population projections as a data source, we calculated the total number of effectively covered young children and women for 10 years (years 2020–2029). The cost per effectively covered child and woman was estimated by dividing the total 10-year cost of each fortification program by the total number of effectively covered children and women, respectively, over 10 years based on each expert group’s zinc reference values. Because the fortification programs are national in scope, only national estimates of cost-effectiveness were generated.

## 3. Results

The national baseline median (P25, P75) usual total zinc intake by children was 3.6 (2.5, 4.9) mg/day, and the median usual absorbable zinc intake was 0.9 (0.7, 1.1) mg/day (Table 2). The respective data for women were 7.6 (5.6, 10.7) mg zinc intake/day and 1.4 (1.0, 1.9) mg absorbable zinc intake/day. Figure 1 and Figure 2 show the distribution of usual total zinc intakes and the distribution of usual absorbable zinc intakes, as estimated based on Miller’s absorption algorithms [23,24], by region for children and women, respectively. Nationally, the mean (±SE) percentage of total dietary zinc intake estimated as absorbable zinc intake was 26.4 ± 7.2% for children and 35.9 ± 13.3% for women.

### 3.1. Effects of Using Alternative Zinc NRVs on Estimated Prevalence of Inadequate Zinc Intake

The prevalence of inadequate dietary zinc intake was directly related to the estimated dietary requirements published by each expert group and, for children, ranged from 10% nationally based on the IZiNCG dietary requirements to 49% nationally based on the EFSA dietary requirements (Figure 3). The prevalence of inadequate intake was intermediate based on the IOM and WHO dietary requirements. This pattern was consistent across all regions. For three of the published physiological requirements for absorbed zinc, the estimated prevalence of inadequate intake was greater than the prevalence based on the dietary requirement, but the reverse was true for the IZiNCG physiological requirement.

As with children, the prevalence of inadequate zinc intake among women was directly related to the estimated dietary requirement. For women, the estimated prevalence of inadequate intake based on the dietary requirement ranged from 14 ± 1.3% (WHO) to 68 ± 0.8% (EFSA) nationally, depending on which expert group’s NRVs were applied (Figure 4). When physiological requirements were applied, the national prevalence of inadequate zinc intake ranged from 9 ± 0.8% (WHO) to 93 ± 0.7% (IOM). The difference in the estimated prevalence of inadequate zinc intake when using the physiological requirement versus the dietary requirement was inconsistent and was particularly large in the North Region, where both the total zinc intakes and phytate intakes were greater than in other regions. When the IOM and IZiNCG physiological requirements were corrected, as described in the Hambidge et al. (2011) study [30], the prevalence of inadequate zinc intake aligned more closely with the EFSA estimates (~86–88%).

### 3.2. Effects of Using Alternative Zinc NRVs on the Predicted Reduction in the Prevalence of Zinc Inadequacy Due to the Presence of Zinc Fortification Programs and Simulated Levels of Fortification

Children: The predicted reduction in inadequate zinc intake following zinc fortification of both wheat flour and bouillon cubes varied substantially depending on the NRVs used (Figure 5, Appendix A) and the corresponding initial prevalence of dietary inadequacy with only wheat flour fortification in place. In almost all cases, the absolute prevalence of inadequacy declined as the level of fortification of bouillon cubes increased, but this decline was greater when the initial prevalence was higher, for example, when based on the EFSA physiological requirements. One exception was observed in Yaoundé/Douala because the proportion of the population consuming fortified wheat flour, and hence the impact of wheat flour fortification on dietary adequacy, was already high. Compared to using the dietary requirement, the reduction in the inadequacy estimated based on the physiological requirement was greater for all expert groups except IZiNCG. The pattern was similar nationally and in the macro-regions for all fortification scenarios tested (i.e., wheat flour, bouillon cubes, and combinations of wheat flour and bouillon cubes).

Women: The initial prevalence of inadequate intake was generally higher for women than children according to most NRVs (Figure 4). As with children, NRVs that resulted in a higher initial prevalence of inadequacy among women and in macro-regions resulted in greater predicted declines in the prevalence of inadequacy as the level of bouillon cube fortification increased (Figure 6, Appendix A). The predicted reduction in the prevalence of inadequate intake, as estimated based on physiological requirements, was greater than when estimated based on dietary requirements for all fortification levels of bouillon cubes and all macro-regions except the South in the presence of wheat flour fortification (Figure 6, Appendix A). In the South macro-region, the predicted reduction in the prevalence of inadequate zinc intake based on the dietary requirement was higher than when estimated based on physiological requirements. In the absence of wheat flour fortification, this pattern was not consistent across the NRVs and macro-regions (Appendix A).

The predicted declines in the prevalence of inadequate zinc intake based on physiological requirements varied substantially among expert groups. For example, bouillon cubes fortified at 5 mg zinc/g bouillon cube were predicted to reduce zinc inadequacy by only 2 percentage points nationally based on the WHO physiological requirement but by 58 percentage points based on the IOM physiological requirement (Appendix A). EFSA and the corrected IZiNCG physiological requirement predicted the same magnitude of decline in zinc inadequacy for all regions and fortification scenarios simulated (Appendix A).

### 3.3. Effects of Using Alternative Zinc NRVs on Cost-Effectiveness of Zinc Fortification Programs

We conducted a cost-effectiveness analysis of the effects of zinc fortification of wheat flour (95 mg zinc/kg flour) and bouillon cubes (5 mg zinc/g bouillon cube) to reduce the prevalence of dietary zinc inadequacy among children and women. The cost-effectiveness of zinc fortification programs varied substantially depending on the predicted effects of zinc fortification on dietary adequacy, as estimated using NRVs from the four expert groups (Table 3). For zinc-fortified wheat flour, the cost per effectively covered child ranged from in US$ 1.05 (EFSA physiological requirement) to US$ 9.45 (IZiNCG physiological requirement) per year. For zinc-fortified bouillon cubes, the cost per child effectively covered ranged from US$ 0.76 (EFSA physiological requirement) to US$ 6.65 (IZiNCG physiological requirement). The cost per effectively covered woman per year from zinc-fortified wheat flour ranged from US$ 0.44 (EFSA physiological requirement) to US$ 24.28 (WHO dietary requirement). For zinc-fortified bouillon cubes, the cost per effectively covered woman per year ranged from US$ 0.27 (IOM physiological requirement) to US$ 6.32 (WHO dietary requirement).

### 3.4. Effects of Using Alternative Zinc UL Values on Prevalence of Zinc Intake above the UL Due to Zinc Fortification Programs

We estimated the effect of zinc fortification on the prevalence of dietary zinc intake above the UL using UL reference values established by WHO, IOM, IZiNCG, and EFSA. It is recommended that the prevalence of intakes greater than the UL not exceed 5% following the introduction of a fortification program [15]. For children, more than 5% nationally consumed more than the IOM, IZiNCG, and EFSA ULs even without any fortification, and 4.2% exceeded the WHO UL (Figure 7). With the introduction of wheat flour fortification, more than one-fourth of children exceeded the ULs published by the three entities other than WHO. At the highest level of hypothetical bouillon cube fortification, the prevalence of intakes above the UL among children barely exceeded 5% for the WHO UL and was ~50% for the other three entities. With fortification of both wheat flour and bouillon cubes, the prevalence of zinc intake above the UL ranged from 9.4% (WHO) to 70% (EFSA).

At the national level, a small proportion of women were predicted to consume more than the UL, both before and after the introduction of wheat flour fortification. At the highest level of bouillon cube fortification, nearly one-fifth of women were estimated to consume more than the EFSA UL, but less than 5% exceeded the ULs published by the other three groups (Figure 8). The prevalence of zinc intake above the UL for each macro-region is reported in Appendix A.

## 4. Discussion

We examined the effects of applying different methodological approaches for developing zinc NRVs and accounting for zinc absorption on the final values published by four different expert groups, and the resulting effects of these widely ranging NRVs on the estimated prevalence of dietary zinc inadequacy among children and women in Cameroon were estimated. We also compared the potential impact of zinc fortification programs on the prevalence of dietary inadequacy and intakes above the UL, as well as program cost-effectiveness, according to the different NRVs. Our analysis shows that the estimated prevalence of dietary zinc inadequacy and the predicted impact and cost-effectiveness of zinc fortification programs vary substantially depending on the NRV applied. In particular, the dietary and physiological requirements for children and the estimated prevalence of inadequate intake were highest for EFSA, followed by WHO, IOM, and IZiNCG. For women, IOM published the highest physiological requirement, followed by EFSA, IZiNCG, and WHO. There are two- to four-fold differences in the estimated prevalence of inadequate intake using the lower EAR versus the highest one. When the corrected IZiNCG and IOM values were used, the prevalence of inadequate intakes was generally consistent with the results based on the EFSA physiological requirement. Within each expert group, the estimated prevalence of dietary zinc inadequacy tended to be higher when calculated based on the physiological requirements, but the results were not consistent across all expert groups. The estimated impact of zinc fortification programs and their cost-effectiveness were positively related to the baseline prevalence of dietary inadequacy. Finally, we found that the prevalence of zinc intake above the UL among children based on IOM, IZiNCG, and EFSA UL values was fairly consistent but >5% for even the lowest level of fortification. By contrast, the prevalence of zinc intake above the UL was consistently < 5% among women, except with the highest level of bouillon cube fortification when using the EFSA UL and in the case of combined wheat flour (95 mg zinc/kg flour) and bouillon cube (5 mg zinc/g bouillon cube) fortification, regardless of which expert group’s UL was applied. The foregoing differences in the estimated prevalence of inadequate zinc intake and effects of zinc fortification have profound implications for justifying and planning zinc fortification programs, as discussed below.

Because of these differences in the estimated prevalence of dietary zinc inadequacy, decisions regarding the need for and the appropriate level of zinc fortification in particular settings would vary depending on the NRVs applied. The IOM dietary requirement and IZiNCG dietary and physiological requirements did not identify zinc inadequacy as a public health concern among children in Cameroon (<25% prevalence of inadequacy) [49]; however, when WHO and EFSA NRVs were used, the prevalence of inadequate zinc intake was greater than the 25% prevalence of dietary inadequacy often applied to identify a public health problem. The challenge, then, is to decide which of the expert group recommendations are more appropriate to apply for these purposes.

With each iteration of the NRVs, the respective expert groups were able to build on the conceptual frameworks of the former groups and benefit from both the earlier decisions and the previously assembled data sources when preparing their recommendations. As such, it seems that the more recent EFSA recommendations and the corrected IOM and IZiNCG values are more likely to approximate the correct values. Of note, the same survey in Cameroon that was used for the present dietary analyses found that the prevalence of low plasma zinc concentration (PZC) ranged from 74% to 92% by macro-region (83% nationally) in children and from 76% to 89% by macro-region (82% nationally) in women [43]. This high prevalence of low PZC is more consistent with the high levels of dietary inadequacy identified by the EFSA (physiological requirement) and corrected IOM and IZiNCG NRVs compared to the earlier ones. Similar comparisons in additional populations and other population subgroups, such as adolescents and adult males, would be helpful to determine the constancy of these findings.

The predicted impact of zinc fortification on the prevalence of inadequate zinc intake was positively correlated with the baseline prevalence of inadequate zinc intake. NRVs that yielded higher estimates of the baseline prevalence of inadequate intake also resulted in a greater predicted impact of zinc fortification. This is because there is more room for improvement when the baseline prevalence of inadequate intake is higher, whereas the potential benefit of fortification is less when the baseline prevalence of inadequate intake is low.

The differences observed in the estimated baseline prevalence of inadequate zinc intake and the predicted impact of fortification programs translated into corresponding differences in the estimated cost-effectiveness of zinc fortification programs. For Cameroon, all four NRVs ranked bouillon cube fortification at 5 mg zinc/g bouillon cube as more cost-effective than wheat flour fortification (95 mg zinc/kg flour), largely because of the greater reach and effective coverage of bouillon cubes. However, the cost of bouillon cube fortification per child-year effectively covered ranged from USD 0.76 to USD 6.65, depending on which NRV was used. A study by Sharieff et al. estimated that the average cost of home fortification for zinc with micronutrient powder was approximately USD 5.4/child/year [50,51]. The more recent zinc NRVs (EFSA and corrected IOM and IZiNCG values) would rate zinc fortification of both bouillon cubes and wheat flour as more cost-effective platforms for delivering zinc than home fortification.

The NRVs that were examined differ with respect to the methods and data sources used to estimate endogenous fecal zinc losses and non-intestinal (urine, integument, semen, and menstrual fluid) zinc losses, as well as methods used to examine the relationship between dietary zinc intake and absorbed zinc [22]. The difference in estimated EFZ losses was mostly responsible for the difference in physiological zinc requirements established by these expert groups. The variation in dietary requirements is directly linked to the variation in physiological zinc requirements as well as the assumed zinc bioavailability from habitual diets stated by these expert groups. The WHO method of estimation of EFZ has been criticized because it was based on only two studies that assessed low-zinc diets without the benefit of tracers, so a somewhat arbitrary figure was used to correct for the effects of the low-zinc diet on fecal and urinary zinc losses and zinc balance [35]. For this reason, the WHO values should be considered less reliable than the more recent ones indicated above, including the corrected IOM and IZiNCG values.

The adjustments applied to account for zinc bioavailability in the dietary requirements represent another source of uncertainty and variation across the expert groups. In this study, the fractional absorption of zinc, as estimated by the algorithm from Miller and colleagues, was ~27% nationally among children, which is similar to the 27–34% applied by IZiNCG for “mixed or refined vegetarian diets” and to the 30% assumed by IOM and EFSA. For women, the estimated fractional zinc absorption of ~35% nationally was consistent with the 34% applied in the “moderate bioavailability” category presented by IZiNCG [22] and EFSA (33%) [37] but lower than that assumed by IOM (48%) [36]. As indicated from tracer studies, the relationship between total zinc intake and absorbed zinc is not linear and is also influenced by the level of phytate in the diet, at least in adults [23]. To account for these effects of the diet on zinc absorption, the dietary requirement should be based on the amount of zinc intake from a specific diet that is needed to replace endogenous losses and provide for tissue accrual. An advantage of applying the algorithms developed by Miller and colleagues is that the estimated fractional zinc absorption is tailored to the zinc (and, for adults, phytate) content of the diets reported in the dataset, rather than assuming a single value for all individuals. However, the method for adults requires the calculation of both total zinc and phytate intakes, and phytate values are not included in all food composition tables, which limits the feasibility of applying this algorithm. For children, the observation that dietary phytate did not predict zinc absorption [24] is inconsistent with other evidence indicating that phytate reduction increased zinc absorption [52,53]; further efforts to understand zinc absorption among young children may improve estimates of absorbable zinc in dietary studies.

WHO fortification guidelines recommend that fortification programs minimize the risk of intake above the UL [15]. More than 5% of the children in the current study were consuming more than the ULs of all of the expert groups except WHO, for which the prevalence was just below 5%, even before the introduction of zinc fortification. This is despite the fact that zinc deficiency was common in this population. Likewise, studies in the USA have found that a sizeable percentage of young children consume more than the IOM UL [54,55], and no harmful effects of these zinc intakes have been reported. This suggests that the safety factors built into the UL may be overly conservative and the UL may be set inappropriately low, which could limit the ability of fortification programs to address the high risk of deficiency in some settings. There are several explanations for these observations. First, the zinc UL is based on detecting a reduction in copper status indicators, regardless of whether these values fall outside the normal range or are associated with any harmful clinical effects. Second, ceruloplasmin, the major copper transporter in plasma, is a positive acute-phase reactant, so any reduction in the prevalence of inflammation associated with higher zinc intakes would be reflected by an apparent lowering of copper status. Available studies assessing copper status in response to zinc did not control for inflammation. Finally, the ULs do not take zinc bioavailability into account, so they overestimate the amount of absorbable zinc in the diet. For all of these reasons, it is likely that all of the current zinc ULs for children are lower than necessary and should be reconsidered.

## 5. Conclusions and Implications

The 2006 WHO/FAO food fortification guidelines recommend using dietary simulation studies to evaluate the potential impact of fortification programs. These simulations are useful to guide policy decisions concerning whether interventions to reduce nutrient inadequacy and the associated consequences are needed and, if so, the most cost-effective strategy [19,56]. However, consensus on NRVs is needed for the dietary data to be interpretable. The present study demonstrates that the estimated prevalence of inadequate zinc intake and the predicted benefit and cost-effectiveness of zinc fortification vary substantially depending on which expert group’s NRVs are applied, and the magnitude of these differences is sufficient to affect policy and program decisions. Thus, these expert groups should revisit the current dietary and physiological requirements for young children and women to achieve harmonized reference values. In the meantime, it seems that the EFSA NRVs and the corrected IOM and IZiNCG values should be applied preferentially to model the impact of food-based zinc interventions on the prevalence of dietary zinc inadequacy.

## Figures and Tables

**Figure 1 nutrients-14-00883-f001:**
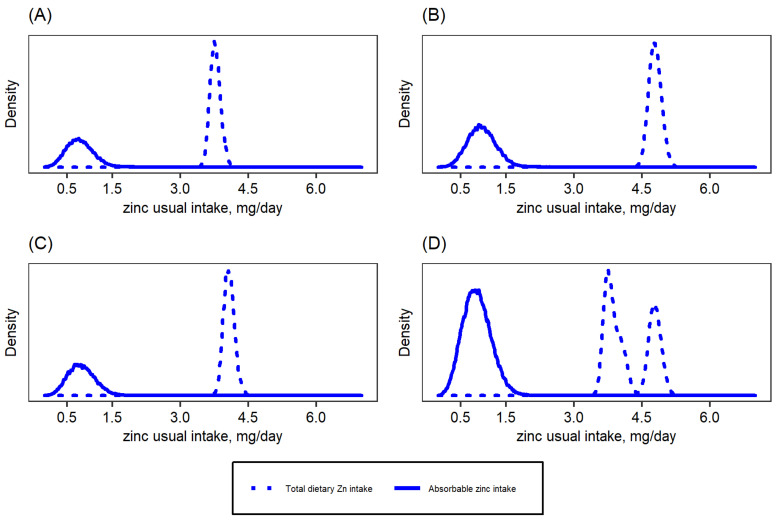
Total dietary and absorbable zinc intake distributions for children aged 12–59 months in Cameroon: (**A**) South macro-region, (**B**) North macro-region, (**C**) Yaoundé/Douala, (**D**) National. Absorbable zinc intake was estimated based on Miller’s equation [24].

**Figure 2 nutrients-14-00883-f002:**
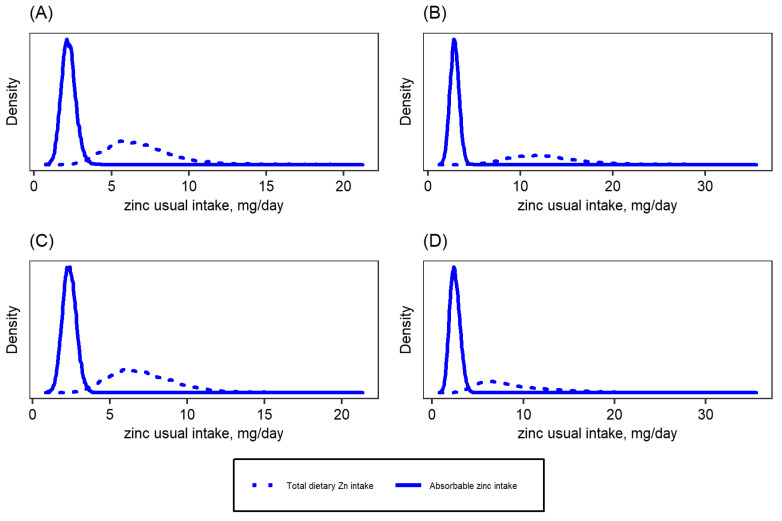
Total dietary and absorbable zinc intake distributions among women of reproductive age in Cameroon: (**A**) South macro-region, (**B**) North macro-region, (**C**) Yaoundé/Douala, (**D**) National. Absorbable zinc intake was estimated based on Miller’s equation [23].

**Figure 3 nutrients-14-00883-f003:**
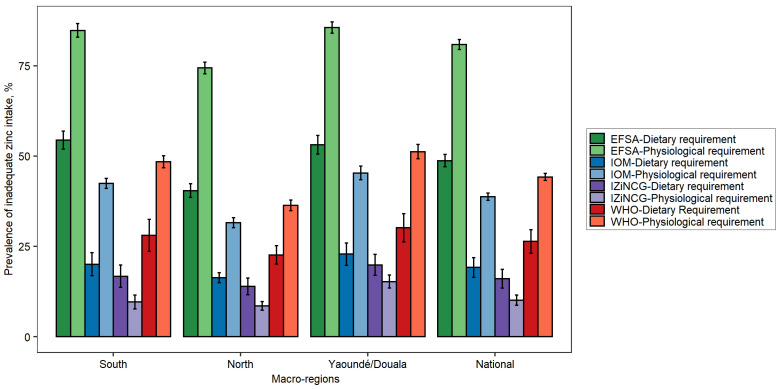
Estimated prevalence of inadequate zinc intake among children aged 12–59 months in Cameroon according to NRVs published by different expert groups. Error bars indicate standard errors. EFSA, European Food Safety Authority; IOM, US Institute of Medicine; IZiNCG, International Zinc Nutrition Consultative Group; WHO, World Health Organization.

**Figure 4 nutrients-14-00883-f004:**
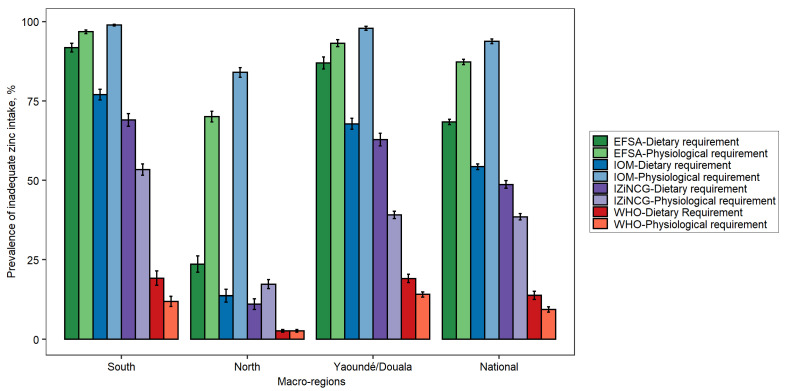
Estimated prevalence of inadequate zinc intake among women of reproductive age in Cameroon according to different NRVs. Error bars indicate standard errors. EFSA, European Food Safety Authority; IOM, US Institute of Medicine; IZiNCG, International Zinc Nutrition Consultative Group; WHO, World Health Organization.

**Figure 5 nutrients-14-00883-f005:**
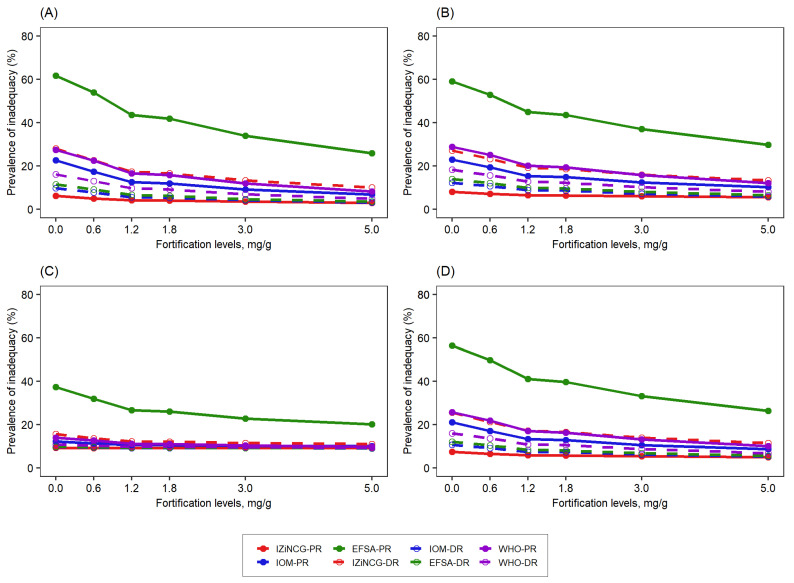
Effect of the level of zinc fortification of bouillon cube (mg zinc/g bouillon cube) on the prevalence of inadequate zinc intake among children in the presence of wheat flour fortification (95 mg zinc/kg flour) according to different NRVs. (**A**) South-macro region, (**B**) North macro-region, (**C**) Yaoundé/Douala, (**D**) National. WHO-PR, World Health Organization—physiological requirement; WHO-DR, World Health Organization—dietary requirement; IOM-PR, Institute of Medicine—physiological requirement; IOM-DR, Institute of Medicine—dietary requirement; IZiNCG-PR, International Zinc Nutrition Consultative Group—physiological requirement; IZiNCG-DR, International Zinc Nutrition Consultative Group—dietary requirement; EFSA-PR, European Food Safety Authority—physiological requirement; EFSA-DR, European Food Safety Authority—dietary requirement.

**Figure 6 nutrients-14-00883-f006:**
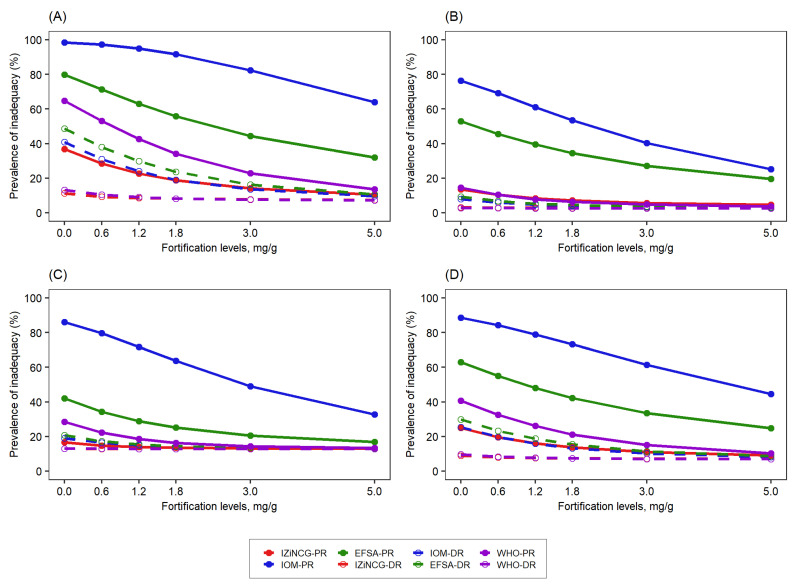
Effect of the level of zinc fortification of bouillon cube (mg zinc/g bouillon cube) on the prevalence of inadequate intake among women in the presence of a wheat flour fortification (95 mg zinc/kg flour) according to different NRVs. (**A**) South-macro region, (**B**) North macro-region, (**C**) Yaoundé/Douala, (**D**) National. WHO-PR, World Health Organization—physiological requirement; WHO-DR, World Health Organization—dietary requirement; IOM-PR, Institute of Medicine—physiological requirement; IOM-DR, Institute of Medicine—dietary requirement; IZiNCG-PR, International Zinc Nutrition Consultative Group—physiological requirement; IZiNCG-DR, International Zinc Nutrition Consultative Group—dietary requirement; EFSA-PR, European Food Safety Authority—physiological requirement; EFSA-DR, European Food Safety Authority—dietary requirement.

**Figure 7 nutrients-14-00883-f007:**
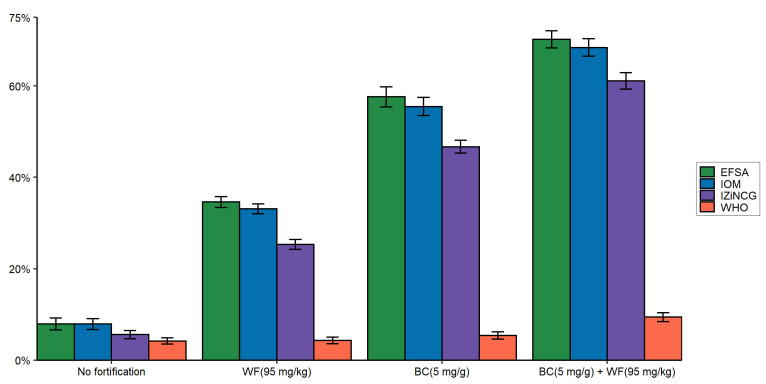
Prevalence (±SE) of zinc intake above the UL, considering the potential contribution of zinc fortification programs among children in Cameroon: estimated based on WHO, IZINCG, IOM, and EFSA UL reference values. EFSA, European Food Safety Authority; IOM, US Institute of Medicine; IZiNCG, International Zinc Nutrition Consultative Group; WHO, World Health Organization; WF, wheat flour; BC, bouillon cubes; UL, tolerable upper intake level.

**Figure 8 nutrients-14-00883-f008:**
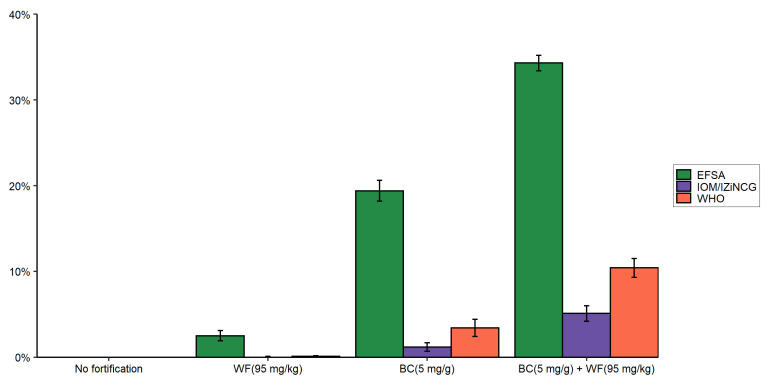
Prevalence (±SE) of zinc intake above the UL considering the contribution of zinc fortification programs among women in Cameroon: estimated based on WHO, IZINCG, IOM, and EFSA UL reference values. EFSA, European Food Safety Authority; IOM, US Institute of Medicine; IZiNCG, International Zinc Nutrition Consultative Group; WHO, World Health Organization; WF, wheat flour; BC, bouillon cubes; UL, tolerable upper intake level.

**Table 1 nutrients-14-00883-t001:** NRVs (mg/day) for young children and women of reproductive age, as developed by the WHO, IOM, IZiNCG, and EFSA ^1^.

Target Group		WHO ^2^			IOM			IZiNCG ^2^			EFSA ^3^			Hambidge et al. Corrected Values [31].	
		PR	DR	UL	PR	DR	UL	PR	DR	UL	PR	DR	UL	IOM-PR	IZINCG-PR
Children	1–3 years	0.83	2.76	23	0.74	2.2	7	0.53	2.0	8 ^a^	1.074	3.6	7	NA	NA
	4–5 years	0.97	3.23	23	1.20	4.0	12	0.83	4.0	14 ^a^	1.390	4.6	10		
Nonlactating, nonpregnant women		0.98	3.25	35	3.30	6.8	40	1.86	7	40	2.9	8.9	25	2.97	2.89
Pregnant women ^4^		1.35	4.5 ^5^	35	3.59	9.5	40	2.56	10	40	3.33	10.2	25	3.36	3.59
Lactating women ^4^		1.88	6.23 ^5^	35	4.55	10.4	40	2.86	8	40	4.03	11.3	25	4.32	3.89

WHO, World Health Organization; IOM, US Institute of Medicine; IZiNCG, International Zinc Nutrition Consultative Group; EFSA, European Food Safety Authority ^1^ PR refers to physiological requirement; DR refers to dietary requirement; NA indicates information not available. ^2^ Dietary requirements for moderately bioavailable diet (diet with average phytate/zinc molar ratio between 5–15 according to WHO and 4–18 according to IZiNCG). ^3^ Dietary requirements for semi-unrefined diet (phytate level: 900 mg/day). ^4^ Physiological requirement for pregnant or lactating women = Requirement for nonpregnant and nonlactating women + requirement increase due to pregnancy or lactation [22]. ^5^ WHO did not publish dietary requirements values for pregnant and lactating women. We calculated the dietary requirement for pregnant and lactating women by dividing their physiological requirement by % zinc absorption assuming moderate bioavailability (the average phytate/zinc molar ratio of diet in Cameroon falls between 5 and 15). ^a^ represents No Observed Adverse Effect Levels (NOAEL).

**Table 2 nutrients-14-00883-t002:** Usual zinc and phytate intakes and phytate/zinc molar ratio among children 12–59 months of age and women of reproductive age in Cameroon.

	South	North	Yaoundé/Douala	National
	Median (P25th, P75th)	Median (P25th, P75th)	Median (P25th, P75th)	Median (P25th, P75th)
Children (N = 860)				
Total dietary zinc intake (mg/day)	3.4 (2.4, 4.3)	4.3 (2.9, 5.8)	3.4 (2.3, 4.5)	3.6 (2.5, 4.9)
Absorbable zinc intake (mg/day)	0.8 (0.6, 1.0)	0.9 (0.7, 1.1)	0.8 (0.6, 1.0)	0.9 (0.7, 1.1)
Estimated fractional zinc absorption (%) ^1^	28.7 (22.1, 33.2)	24.2 (19.3, 30.7)	27.3 (22.2, 31.3)	27.0 (21.2, 31.8)
Phytate intake (mg/day)	357 (197, 570)	460 (261, 724)	331 (160, 545)	390 (213, 622)
Phytate: zinc molar ratio ^2^	13.1 (8.9, 17.6)	12.8 (8.9, 16.4)	10.0 (7.7, 13.8)	11.9 (8.3, 16.0)
Women (N = 902)				
Total dietary zinc intake (mg/day)	6.3 (5.0, 7.8)	12.1 (9.8, 14.6)	6.5 (4.9, 8.1)	7.6 (5.6, 10.7)
Absorbable zinc intake (mg/day)	2.2 (1.8, 2.5)	2.9 (2.58, 3.2)	2.3 (1.9, 2.6)	2.4 (2.0, 2.8)
Estimated fractional zinc absorption (%) ^1^	37.5 (30.0, 49.2)	25.9 (19.9, 33.7)	41.1 (32.2, 49.4)	34.8 (25.2, 45.3)
Phytate intake (mg/day)	772 (567, 1008)	1353 (1058, 1695)	661 (466, 878)	889 (620, 1253)
Phytate/zinc molar ratio ^2^	12.8 (8.7, 17.3)	12.4 (8.4, 15.3)	9.8 (7.2, 14.5)	11.6 (8.1, 15.9)

^1^ $\mathrm{Fractional}\;\mathrm{zinc}\;\mathrm{absorption}\;\left(\%\right)=\frac{\mathrm{absorbable}\;\mathrm{zinc}\;\mathrm{intake}}{\mathrm{total}\;\mathrm{zinc}\;\mathrm{intake}}*100$; ^2^ $\mathrm{Phytate}\;\mathrm{zinc}\;\mathrm{molar}\;\mathrm{ratio}=\frac{\mathrm{mg}\;\mathrm{phytate}\;\mathrm{per}\;\mathrm{day}/660}{\mathrm{mg}\;\mathrm{zinc}\;\mathrm{per}\;\mathrm{day}/65.4}$.

**Table 3 nutrients-14-00883-t003:** Cost-effectiveness of 10-year zinc fortification programs in addressing the prevalence of inadequate zinc intake in Cameroon: estimated based on WHO, IOM, IZINCG, and EFSA dietary requirements and physiological requirements. The cost is in US$.

Reference Values Applied		Cost Per Effectively Covered Child Per Year		Cost Per Effectively Covered Woman Per Year	
		Wheat Flour (95 mg/kg)	Bouillon Cube (5 mg/g)	Wheat Flour (95 mg/kg)	Bouillon Cube (5 mg/g)
WHO	Dietary requirement	2.48	1.84	3.03	2.26
	Physiological requirement	1.36	1.13	24.28	6.32
IOM	Dietary requirement	3.65	2.60	0.50	0.35
	Physiological requirement	1.44	1.19	0.61	0.27
IZiNCG	Dietary requirement	4.73	3.23	0.52	0.39
	Physiological requirement	9.45	6.65	0.89	0.54
EFSA	Dietary requirement	1.11	0.93	0.44	0.28
	Physiological requirement	1.05	0.76	0.50	0.28

EFSA, European Food Safety Authority; IOM, US Institute of Medicine; IZiNCG, International Zinc Nutrition Consultative Group; WHO, World Health Organization.

## Data Availability

Data will be made available on request to the authors, subject to the data sharing conditions of the ethical review boards with oversight for the study.

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
