# Peer review of "Applying Zinc Nutrient Reference Values as Proposed by Different Authorities Results in Large Differences in the Estimated Prevalence of Inadequate Zinc Intake by Young Children and Women and in Cameroon"

_nutrients, 2022, doi:10.3390/nu14040883_

Round 1
Reviewer 1 Report
dear authors, thank you for this interesting study. the major problem of the study is sample as it included children and women only and children only <6 years thus due to the gap in age maybe the paper need to be split into two and focus on age reproductive women.
some additional changes to consider.
title: need to indicate that this is limited to children and women only and consider shortening - see my comment above
abstract: reported prevalence need 95%ci we like to see the overlap between different nrvs
Table 1 needs to be linked with formulating the problem. the introduction could have focused on the approach of developing the nrvs by each authority.
Give the eligibility criteria, and the sources and methods of selection of participants e.g. did you include people with comorbidities its not very clear we need to see Table 1 for the participants too
Explain how the study size was arrived at clearly the sample is not balanced 860 children and 902 women (women have 4 categories and need to know each N)
Describe any sensitivity analyses - please make underlying data and analysis codes available.
Discuss the generalisability (external validity) of the study results given you missed teenagers and youth especially female ones.
Author Response
REVIEWER # 1
Comment #1: dear authors, thank you for this interesting study. the major problem of the study is sample as it included children and women only and children only <6 years thus due to the gap in age maybe the paper need to be split into two and focus on age reproductive women.
- We appreciate the reviewer’s comment, but respectfully disagree for several reasons. First, young children and women of reproductive are of particular concern both because of their relatively high requirements for growth and reproduction and the existing evidence of functional derangements due to zinc deficiency in these population sub-groups. Second, by analyzing the effects of using NRVs published by four different expert committees in both of these population sub-groups we are able to compare whether they produce consistent effects. Finally, because individual-level dietary intake data are typically more available for these population sub-groups we were able to assess outcomes using empirical data. For these reasons, we believe that the focus on young children and WRA is appropriate.
some additional changes to consider.
Comment #2: title: need to indicate that this is limited to children and women only and consider shortening - see my comment above
- We agree that the title is long, but we considered this to be justified for the title to be clear, indicative of the conclusion and complete. As suggested, we added the target group in the revised title and reduced the word count. The current revised title is “Applying zinc nutrient reference values as proposed by different authorities results in large differences in the estimated prevalence of inadequate zinc intake by young children and women in Cameroon”.
Comment #3: abstract: reported prevalence needs 95%ci we like to see the overlap between different nrvs
- We used standard errors to estimate uncertainties as recommended by the National Cancer Institute methods. We calculated the 95% CI from the standard errors. We added the 95% CI in the current version of the abstract.
Comment #4: Table 1 needs to be linked with formulating the problem. the introduction could have focused on the approach of developing the nrvs by each authority.
- We believe that the detailed description of each expert group’s approach to determining zinc requirements is more appropriately placed in the Methods section (section 2.2.1), so as not to expand the length of the Introduction. In response to the reviewer’s suggestion, we have added a note to the Introduction to indicate that the detailed description of each expert group’s approach is provided below (see line # 85 of revised paper). If, however, the Editor would prefer that we move this information to the Introduction, we are willing to do so.
Comment #5: Give the eligibility criteria, and the sources and methods of selection of participants e.g. did you include people with comorbidities its not very clear we need to see Table 1 for the participants too
- We have added information on the study inclusion and exclusion criteria, including reference to our previously reported paper (line # 126, reference # 25).
Comment #6: Explain how the study size was arrived at clearly the sample is not balanced 860 children and 902 women (women have 4 categories and need to know each N).
- The original sample size was calculated to be representative of women in reproductive age and children age 12-59 months in Cameroon and at the “macro-regional” (stratum) level, for the purpose of evaluating a national food fortification program. As described below, there are several reasons why we had a smaller sample size for children than women.
- We excluded a few observations that had implausible energy intakes (+ 3 SD above the mean energy intake). A few more children than women were excluded by this criterion.
- We excluded 14 child-days from the analysis who inflated the within to between-person variance ratio (> 10) as recommended by Davis et al 2019. For women, the within-person to between-person variance ratio was in the acceptable range (< 10), so no woman was excluded by this criterion.
- In addition, attrition between recruitment and data collection was slightly greater among children compared to women.
- We cited references (reference # 29 and 30) that provide a detailed description of the inclusion and exclusion criteria, sample size and sampling approach (lines 134-136).
Comment #7: Describe any sensitivity analyses - please make underlying data and analysis codes available.
- The focus of the manuscript could be considered a sensitivity analysis of how the outcomes vary depending on the NRV and adjustment for absorption applied. We did not conduct further sensitivity analyses beyond this stated objective. Consistent with policies of the Gates Foundation, which supported this analysis in part, the codes and the data will be available based on a reasonable request.
Comment #8: Discuss the generalisability (external validity) of the study results given you missed teenagers and youth especially female ones.
- As indicated above, we focused on women and young children for several reasons. As suggested by the reviewer, we have added a sentence to the Discussion (lines 591-2) stating that it would be of interest to examine the effects of different NRVs on the predicted prevalence of inadequate intakes by other population sub-groups. Thus, the results do not directly apply to other age groups such as adolescents. However, the focus on specific target groups does not change the conclusion that NRVs should be reconsidered and harmonized.
Reviewer 2 Report
Dear authors,
The paper "Applying zinc nutrient reference values as proposed by different authorities results in large differences in the estimated prevalence of inadequate zinc intake and predicted impact and cost effectiveness of zinc fortification programs" is a very important socio-economic issue in terms of population health.
The presented work is a very important contribution to further discussion on standardization of reference values not only for zinc but also for other essential elements.
As it is known, in the case of zinc its chemical form plays an important role in its bioavailability. I propose to expand this aspect in the paper.
Moreover:
- in Table 1, footnote 4 should be moved (aligned with others).
- In Figures 5 and 6 in the legend, WHO-DR appears twice, and the description of the figure lacks the explanation of the acronyms EFSA-PH and EFSA-DR.
Regards
Author Response
REVIEWER # 2
Dear authors,
The paper "Applying zinc nutrient reference values as proposed by different authorities results in large differences in the estimated prevalence of inadequate zinc intake and predicted impact and cost effectiveness of zinc fortification programs" is a very important socio-economic issue in terms of population health.
- Thanks for your compliment
Comment #1: The presented work is a very important contribution to further discussion on standardization of reference values not only for zinc but also for other essential elements.
As it is known, in the case of zinc its chemical form plays an important role in its bioavailability. I propose to expand this aspect in the paper.
- We agree that bioavailability is an important consideration. Each of the expert groups had its own specific approach of considering zinc bioavailability as explained in the method section and supplementary table 1 of the manuscript. In section 2.2.2, we briefly described the approach applied to estimate bioavailable (absorbable) zinc intake in this manuscript, and we discussed the implications of the bioavailability assessment in the discussion (lines 627- 632). For the data analysis, we classified the diet in this study population as moderately bioavailable diet based on the observed phytate: zinc molar ratio.
Comment #2: Moreover: - in Table 1, footnote 4 should be moved (aligned with others).
- Thank you! Corrected accordingly
Comment #3: In Figures 5 and 6 in the legend, WHO-DR appears twice, and the description of the figure lacks the explanation of the acronyms EFSA-PH and EFSA-DR.
Thanks! Corrected accordingly. We revised Figures 5 and 6 legend labels
Round 2
Reviewer 1 Report
dear authors, the comments were not addressed adequately.
comments 5,6,7 are important sources of bias.
Author Response
Comment # 1: dear authors, the comments were not addressed adequately. comments 5,6,7 are important sources of bias.
Response: We understand the concern regarding the potential bias that could be introduced into a study by the eligibility criteria and attrition. For this secondary analysis, we addressed this issue by reporting the inclusion and exclusion criteria of the original survey and referring readers to earlier publications that include the details of survey sampling. With this information, interested readers may understand in detail the representativeness and potential sources of bias in sampling and eligibility assessment.
In addition, as we mentioned in our previous comments, there were several reasons for excluding observations from the dataset prior to analysis, including 1) those with implausible single-day energy intakes and 2) observations that influenced the within-to-between person variance ratio, the most important factor influencing the shape of the estimated distribution of usual dietary intakes. Thus, exclusion of these observations provides a more robust estimate of the prevalence of inadequate intake in this dataset. Notably, the total number of children excluded from analysis is about 5% of the total sample size, which is unlikely to greatly affect the composition of the sample.
Finally, we note that issues related to representativeness of the sample do not affect the main conclusion of the paper, which relate to comparing the use of different cutoffs/method applied in the same dataset. That is, although the prevalence of inadequate intake will depend on the specific dataset/sample, the within-dataset comparison of the effects of use of different cutoffs and absorption adjustments is valid and useful.